# New Targets and Strategies for Rheumatoid Arthritis: From Signal Transduction to Epigenetic Aspect

**DOI:** 10.3390/biom13050766

**Published:** 2023-04-28

**Authors:** Menglin Zhu, Qian Ding, Zhongxiao Lin, Rong Fu, Fuyuan Zhang, Zhaoyi Li, Mei Zhang, Yizhun Zhu

**Affiliations:** 1State Key Laboratory of Quality Research in Chinese Medicine, School of Pharmacy, Macau University of Science and Technology, Macau 999078, China; 2Shanghai Key Laboratory of Bioactive Small Molecules, Department of Pharmacology, School of Pharmacy, Fudan University, Shanghai 201203, China

**Keywords:** rheumatoid arthritis, molecular mechanisms, epigenetic

## Abstract

Rheumatoid arthritis (RA) is a chronic autoimmune disease that can lead to joint damage and even permanent disability, seriously affecting patients’ quality of life. At present, the complete cure for RA is not achievable, only to relieve the symptoms to reduce the pain of patients. Factors such as environment, genes, and sex can induce RA. Presently, non-steroidal anti-inflammatory drugs, DRMADs, and glucocorticoids are commonly used in treating RA. In recent years, some biological agents have also been applied in clinical practice, but most have side effects. Therefore, finding new mechanisms and targets for treating RA is necessary. This review summarizes some potential targets discovered from the perspective of epigenetics and RA mechanisms.

## 1. Introduction

Rheumatoid arthritis (RA) is a chronic autoimmune disease that affects the joints and can lead to inflammation, pain, stiffness, and eventually, joint damage and disability. According to statistics, not only does the prevalence of rheumatoid arthritis vary widely among ethnic groups, but it is more common in women than in men. Lifestyle habits can also affect the incidence of RA. The impact of smoking on RA is enormous. Studies have shown that people who have smoked for 20 years are twice as likely to develop RA as people who do not smoke [1]. Relevant data also support the impact of nutritional habits on RA. Studies have shown that overeating food containing saturated fat, trans fat, high salt, and high calories will promote inflammation and induce insulin resistance, increasing the risk of RA [2,3]. Studies have shown that people with RA have different gut microbiomes compared to those without RA and that certain bacteria in the gut can affect inflammation and joint damage [4]. Data have shown that the diversity of intestinal flora in RA patients is lower than in patients without RA.

In the early stage of RA, the immune system is activated and inflammatory cells, such as macrophages, T cells, and B cells, infiltrate the synovial tissue of the affected joint. With the intensification of immune cell penetration, RA gradually becomes severe, eventually leading to hyperplasia of the lining layer and pannus formation. The healthy synovial tissue mainly contains macrophage-like synovial (MLS) and fibroblast-like synovial (FLS), while the synovial tissue of RA was heavily infiltrated with leukocytes [5]. Synovial inflammation in RA is maintained by interactions between T cells, macrophages, and FLS. In the joint, once the white blood cells are recruited and activated, they cannot be stopped, thus causing inflammation and causing damage to the joint [6]. Studies have shown that non-classical monocytes play an essential role in RA [7]. During an RA flare-up, neutrophils are activated and migrate to the joints, where they release inflammatory molecules that contribute to the destruction of joint tissue [8]. In the inflammatory milieu of the joints, it is theorized that the recruited monocytes differentiate into macrophages and produce inflammatory mediators, thus activating the immune system and inducing RA [9]. The abnormal proliferation and invasion of FLS caused by the secretion of pro-inflammatory factors play an important role in joint injury and inflammation. We summarize the structure of the joint and the changes in rheumatoid arthritis in Figure 1.

The development of RA: Under the stimulation of susceptibility factors, the adaptive immune system will recognize peptidyl arginine and peptidyl citrulline. Antigen-presenting cells (APCs) will transmit these changes to the lymphatic tissues to generate an immune response and autoantibodies. Then, FLS, APCs, and macrophages are activated to produce inflammatory cytokines (IL-6, 8, 17, etc.). Forming of the immune complex and activating of the complement will stimulate the immune system for a second time, increasing the production of cytokines and synovial vascular inflammation, eventually leading to synovial inflammation and bone destruction [10].

RA is characterized by inflammation of the synovial membrane, the proliferation of synovial cells and the formation of pannus, leading to the dysfunction of the synovial. At present, the treatment of RA is mainly to relieve pain, inhibit the proliferation of synovial cells, prevent the generation of pannus, regulate immune function and protect cartilage, and finally, achieve the goal of eliminating inflammation as far as possible [11,12]. Common drugs include non-steroidal anti-inflammatory drugs (NSAIDs), glucocorticoids, and DMARDs (Disease-modifying anti-rheumatic drugs). NSAIDs’ primary mechanism of action is to inhibit cyclooxygenase activity from combating inflammation. NSAIDs are effective in the initial stages of RA and can be used in combination with DMARDs. However, long-term use of NSAIDs can cause adverse effects on the gastrointestinal tract and kidneys. A variety of cells are involved in the inflammatory response in RA. The effect of glucocorticoid is to relieve the pain and swelling of RA patients by inhibiting the activity of immune cells, including T cells and B cells, which play a role in the autoimmune response that leads to RA [13]. Glucocorticoids are divided into short-term (cortisone and cortisol), medium-term (methylprednisolone, prednisolone, methylprednisolone, and triamcinolone), and long-term (dexamethasone and betamethasone) treatments of RA [14]. In DMARDs, methotrexate (MTX) is often used in combination with a glucocorticoid as the primary drug for RA treatment due to its advantages, such as reliability, effectiveness, low cost, and few side effects. In addition, hydroxychloroquine, sulfasalazine, leflunomide, etc., can also be used in the treatment of RA.

Although there are many treatment options for RA, a large number of patients do not achieve remission, thus indicating the need for a more diverse approach to treatment. The pathogenesis of RA is complex and involves multiple factors, including genetic and epigenetic factors. Signal transduction pathways play a critical role in the pathogenesis of RA. The abnormal activation of various signaling pathways contributes to the production of proinflammatory cytokines, chemokines, and other inflammatory mediators. Epigenetic modifications, including DNA methylation, histone modifications, and noncoding RNA-mediated gene regulation, are also involved in the development and progression of RA. Epigenetic changes can alter gene expression patterns and contribute to the aberrant activation of signaling pathways and the dysregulation of immune responses. Aberrant signals in signal transduction and epigenetic modification often serve as targets for drug discovery [15]. In recent decades, the development of biologically targeted therapy has been very rapid and has provided us with novel ideas. Below, we will review the potential therapeutic targets for RA and provide a reference for subsequent research.

## 2. Cellular Mechanisms in RA

RA is a systemic immune disease causing synovial inflammation, joint swelling, and injury [16]. In the development of RA, the synovial membrane gradually becomes invasive, destroying cartilage and bone. The synovial intimal lining is composed of MLSs and FLSs in equal proportions. Usually, the synovial intimal lining is a thin and loose membrane-like structure with only one or two layers of cells. In comparison, RA intimal lining can be thick with 10–15 layers of cells [17]. MLSs and FLSs cells play an essential role in the development of RA. They mainly secrete a variety of cytokines to stimulate the occurrence of inflammation. In addition, the recruitment of macrophages, T cells, and B cells expands the synovial sublayer and promotes inflammation. The cellular mechanism in RA is shown in Figure 2.

### 2.1. Fibroblast-Like Synovial (FLSs)

In healthy synovial tissue, FLSs promote synovial fluid production, thus protecting cartilage. FLSs also maintain synovial ECM (extracellular matrix) by producing matrix components and ECM-degrading enzymes [18]. FLSs have beneficial effects on healthy tissues, but they also “commit indelible crimes” in the pathogenesis of RA. The source of the destruction of bone tissue and cartilage tissue is FLS-mediated. In RA, different immune cells produce a variety of cytokines and growth factors, leading to the production of inflammation and proliferation of FLSs, leading to the formation of pannus [18]. In addition, FLSs can also resist endoplasmic reticulum stress-induced apoptosis [19]. Metabolic changes of FLSs are also a feature of RA. Studies have shown that changes in the glucose metabolism pathway, lipid metabolism, and amino acid metabolism of RA FLSs can induce synovial hyperplasia and increase inflammation. Increased glycolysis in FLS was also a feature in determining RA. The Hypoxia-inducible factor 1α (HIF1α) is a transcription factor that can induce glycolysis, so targeting HIF1α has also been used as a treatment strategy for RA in recent years [20]. FLSs activate T cells and B cells by producing a variety of cytokines [21] and secreting various enzymes (MMP1, MMP13, MMP3, ADAMTS4, ADAMTS5, and so on) that promote the invasion of immune cells and aggravate inflammation [22].

### 2.2. Macrophage-Like Synovial (MLSs)

MLSs co-exist with FLSs in the synovium, but the function of MLSs is less diverse than that of FLSs [23]. There are not many relevant studies on MLSs, but MLSs are known to induce an inflammatory response during the development of RA [24]. MLS promotes RA development by secreting a variety of factors (oxygen species, nitric oxide intermediates, and matrix-degrading enzymes) [25] and can release a variety of cytokines to stimulate immune cells and FLSs [26].

### 2.3. T Cells

The association of human leukocyte antigen with RA patients reminds us of the important role of T cells in inducing immune responses. In general, type 1 T helper (Th1) and type 17 T helper (Th17) are thought to play a significant role in the pathogenesis of RA, but type 22 T helper cells (Th22) have also recently been shown to promote the progression of RA [27]. T cells were the primary immune infiltrating cells in RA patients. The important role of cytokines in the pathogenesis of RA has been extensively described [28]. Th1 cells and Th17 cells which can produce IL-1 and IL-17 have been proven to cause RA [29]. The role of IL-17 is highly valued in RA because it can induce a variety of cells (synovial cells, macrophages, etc.) to produce a large number of inflammatory factors (IL-1β, IL-6, TNFα), and it plays an important role in the production of chemokines (CXCL1, CXCL2, CXCL8, CCL2, CCL7) [30]. Treg cells proliferate in an inflammatory environment. In the joints and synovium of RA patients, the capacity of Treg cells is reduced. Forkhead box p3 (Foxp3), as an upstream regulator of Treg cells, can attenuate the inhibitory function of Treg cells under the induction of IL-1β and IL-6 [31,32]. Overall, T cells in RA (Th1, Th17, Treg, and Th22) have a complex interaction that leads to increased inflammation [27].

### 2.4. B Cells

The autoantibody production, antigen presentation, and cytokine secretion of B cells affect the progression of RA. There are various subsets of B cells, but data suggest that double negative (CD27^−^IgD^−^) and class-switched memory (CD27^+^IgD^−^) are the key to RA pathogenesis [33]. This may be due to the fact that CD27^+^IgD^−^ is more likely to express RANKL, which induces RA, and in CD27^−^IgD^−^, miR-155 level increased, which is required for the production of autoantibodies by B cells [34,35]. CD21^−/low^ CD27^−^ IgD^−^ B cells have been detected in the peripheral blood and synovial fluid of RA patients, and the cells secrete RANKL even in the absence of stimulation, leading to bone destruction. Therefore, CD21^−/low^ may be a potential target cell for the treatment of RA [36]. B cells act as antigen-presenting cells that can deliver antigens to CD4^+^ helper T cells, triggering an immune response. In RA, B cells play an increasingly important role in activating autoreactive T cells as inflammation worsens. In addition, in the synovial environment of RA patients, inflammatory factors and cytokines (TNF-α, IFN-γ, IL-6, IL-1β, IL-17, CCL20, RANKL, etc.) secreted by B cells promote bone destruction and inhibit bone formation. It also affects T cells and other cells in the synovial membrane, speeding up the progression of the inflammation [37,38,39]. However, the role of Regulatory B (Breg) cells in RA is opposite to that of B cells, which have an immunosuppressive effect and produce anti-inflammatory factors [40]. In addition, B cells are activated and produce autoantibodies in RA.

## 3. Signaling Pathways in RA

The development process of RA is associated with a variety of pathways, such as the NF-κB, MAPK, Wnt, JAK-STAT, PI3K-Akt, and so on. We have recently made a detailed review of the association between pathways in RA [15]. These pathways will be briefly introduced below and summarized in Figure 3. The main aim is to find new targets for the treatment of RA through these pathways.

NF-κB is one of the most common pathways in RA and is often associated with inflammatory responses. NF-κB is a dimer transcription factor associated with the production of several pro-inflammatory factors (TNF-α, IL-1β, and IL-6) in RA [41]. The production of ROS indirectly induces the activation of IkB kinase (IkK), resulting in the degradation of IkB and the induction of inflammation in the nucleus of NF-κB. An obvious sign of RA occurrence is the activation of NF-κB. NF-κB can cause synovial inflammation by prompting macrophages to release inflammatory factors. Many studies have found that the inhibition of the NF-κB signaling pathway can relieve inflammatory symptoms in rats with arthritis. AP1 can regulate cytokines, affect MMPs, NF-κB production, and synovial hyperplasia, and its expression is related to the severity of the disease [42]. ROS can activate the AP1 pathway, leading to the production of pro-inflammatory factors. In IL-18-induced FLS inflammation, Roflumilast can relieve the degree of oxidative stress in cells, reduce the expression of inflammatory factors, and inhibit the activation of NF-κB and AP1 [43]. IL-1β induces ROS production and AP1 activation in RASFs. cPLA2 has been shown to contribute to the development of arthritis. Activated AP1 binds to the promoter region of cPLA2, resulting in increased cPLA2 expression. While ho-1, which has antioxidant effects, attenuates AP-1 and cPLA2 expression, the antioxidant HO-1 can inhibit both AP1 and cPLA2 [43].

The MAPK family includes p38 kinases, ERK, and JNK, which play an important role in inflammation. Phosphorylated forms of MAPKs can induce the transcription and activation of cytokines in RA. ERK is involved in pannus formation, JNK regulates collagenase production in SFs, and p38 regulates MMP3 expression in fibroblasts [44]. MAPKs are closely related to the proliferation and apoptosis of FLS in RA and can promote the proliferation of FLS by activating the JAK-STAT pathway. Hydrogen, as a molecule found in nature, has been shown to neutralize OH and ONOO-. In CIA-induced arthritis, hydrogen was shown to reduce joint swelling in mice. In in vitro experiments, hydrogen can inhibit the proliferation of FLS and has an antioxidant effect, which is accompanied by the activation and inhibition of MAPK signaling pathways [45]. Sonic Hedgehog (SHH) signaling pathway is related to cell proliferation and migration, and its expression is increased in RA synovial tissues and RAFLS [46,47]. Further study found that SHH could activate an intracellular MAPK/ERK cascade [48].

The Wnt pathway plays a role in a variety of cancers and immune diseases and is related to various physiological processes, including cell proliferation and migration [49,50]. It mainly contains two pathways: β-catenin-dependent and β-catenin-independent [51]. It is well known that the Wnt pathway is highly expressed in synovial tissues of RA patients [52]. In addition, the Wnt//β-catenin pathway can promote the generation of bone cells and plays a positive role in human bone development [53]. In RA, the Wnt inhibitor DKK1 can promote pannus formation [49]. Additionally, a variety of miRNAs have been confirmed to regulate the pathogenesis of RA through the Wnt pathway. A recent study by our group shows that the transcription factor E2F1 can bind to the promoter region of Neuron navigator 2 (NAV2), activate the transcription of NAV2 [53], and regulate RA through the Wnt/β-catenin signaling pathway [53], and subsequent studies have found that NAV2 can regulate RA through SSH1L/Cofilin-1 signaling pathway [54]. This provides a new pathway target for the treatment of RA.

The Janus kinases (JAK) family includes Jak1, Jak2, Jak3, and tyrosine kinase (TYK) 2 [55]. STAT family includes STAT1, STAT2, STAT3, STAT4, STAT5a, STAT5b, and STAT6 and is involved in a variety of physiological processes in cells (proliferation, apoptosis, etc.) [56]. The JAK/stat pathway is associated with a variety of cytokines, and when the associated cytokines bind to their receptors, it means that the JAK/STAT signaling pathway is about to activate [56]. When activated, members of the JAK family phosphorylate some tyrosine residues to dock with signaling molecules [55]. STAT members are recruited to the JAK dimer and phosphorylated, separated from the receptor, translocated to the nucleus, and regulated the transcription of genes involved in cell physiological processes [57,58]. Additional good news for JAK inhibitors comes from a recent clinical study that found relief of pain in RA patients with JNK inhibitors [59].

Phosphatidylinositol 3 kinase (PI3K)-Akt plays an important role in cell physiology, regulating cell differentiation, proliferation, autophagy, and so on [60]. PI3K is a heterodimer composed of a p110 catalytic subunit and a p85 regulatory subunit [61]. PI3K can be activated by dimer conformation changes or by binding of Ras and p110. PI3K phosphorylates PIP2 to PIP3, thus binding with a variety of downstream effector molecules and affecting the physiological process of cells [62]. Protein kinase B (AKT) is an important downstream target in the PI3K pathway. After PIP3 recruits PDK1 and AKT to the cell membrane, PDK1 phosphorylates AKT1 to further activate downstream substrates (protein kinases, E3 ubiquitin ligases, regulators of small G proteins, metabolic enzymes, and transcription factors) [63]. In RA, the PI3K-AKT signaling pathway is associated with the expression of a variety of cytokines and participates in various pathological processes, such as pannus formation, cell proliferation, and migration [64]. Abnormal activation of PI3K/AKT pathway also stimulates the expression of HIF-1α and promotes angiogenesis. HIF-1 is involved in various cellular processes, such as glycolysis, cell growth, migration, and apoptosis. HIF-1 activates inflammatory cells and participates in pannus formation. In RA, ubiquitination is inhibited, leading to the accumulation of HIF-1α, which binds to the HRE promoter and regulates the expression of VEGF, ultimately leading to angiogenesis [65].

Based on different cellular mechanisms and targeting pathways in RA, a variety of target inhibitors have been applied in clinical practice (Table 1), but most of them have adverse reactions, even so, JAK inhibitors are still the pinup targets for RA drug development, and some new JAK inhibitors are in clinical trials. Meanwhile, many studies have focused on the formulation of new drugs that can alleviate RA. The extraction of active ingredients from plants has been a research hotspot in recent years, and this paper will summarize recent findings in Table 2.

## 4. Epigenetic Regulation in RA

The word ‘epigenetics’ was first used to describe heritable changes in gene function that occur without a change in the genetic coding. Nowadays, the word is used to characterize chromosomal structural changes that do not entail nucleotide sequence changes, regardless of whether the changes are strictly heritable [105]. DNA methylation and covalent histone modifications are examples of epigenetic changes. Histone modifications include (de)acetylation, (de)methylation, ubiquitination, and sumoylation. These changes control how accessible DNA is to transcription factors. An increasing amount of epigenetic enzymes are discovered to act as ‘writers’ or ‘erasers’ in the modifications of DNA and histones by attaching or removing certain functional groups, resulting in the regulation of gene expression. In addition, some proteins (‘readers’) that are able to recognize epigenetically modified sites also act as key players in the regulation of gene expression. We describe a schematic of epigenetics in Figure 4.

Epigenetics regulate the different levels of DNA and chromatin that accumulate in the nucleus of eukaryotic cells. DNA regulatory regions (such as promoters and enhancers) influence gene transcription by regulating the expression of transcription factors (TF), which is related to the state of chromatin structure. Euchromatin promotes the interaction between DNA-binding proteins and TF and increases gene transcription activity, while heterochromatin does the opposite [106]. Epigenetic modification mainly includes histone modification, DNA methylation, and non-coding RNA modification. Next, this paper will introduce the impact of RA from these three aspects. Potential targets are summarized in Table 3.

### 4.1. Histone Modification

Recent research has shown that histone modifications may be involved in the development and progression of RA. Histones are proteins that help package and organize DNA in the nucleus of cells, and their modification can affect gene expression and protein production. Histones can be modified by methylation, acetylation, phosphorylation, and poly ADP-ribosylation. Histone methylation, which occurs mainly in proteins containing SET domains, is the most stable modification method, and the modification sites are lysine and arginine. Studies have found that specific histone modifications, such as histone acetylation and methylation, are associated with increased inflammation and joint destruction in RA.

Our team found that in PDGF-induced FLS, the expression of Jumonji C family of histone demethylases (JMJD3) was increased through the Akt signaling pathway, and the proliferation and migration ability of FLS was weakened after inhibition or silence of JMJD3, and the symptoms of DBA/1 mice by collagen-induced arthritis (CIA) were alleviated [107]. At the same time, we also found that CSE/H2S can reduce the expression of JMJD3 by inhibiting transcription factor Sp-1 and alleviating arthritis [108]. GATA4 plays an important role in regulating cardiac function. Our team found that GATA4 was up-regulated in MH7A stimulated by IL-1β and could regulate the proliferation and migration of endothelial cells and blood vessel formation through the MAPK signaling pathway [109]. This conclusion may provide ideas for RA treatment.

Some studies have found that the downregulation of SIRT3 can promote the increase in ROS levels in mice and regulate oxidative stress [110]. After the reduction of the SIRT3 level, mitochondrial oxidative stress was aggravated, and osteoarthritis was induced in mice [111]. The role of SIRT4 in RA has not been studied, but SIRT4 can affect osteoarthritis through a variety of inflammatory factors, such as TNFα and IL-6, and the effect of SIRT4 on oxidative stress has also been proven the increased expression of SIRT4 can up-regulate SOD1, SOD2, and catalase [112].

Recent research has suggested that histone deacetylases (HDACs) may be involved in the development and progression of RA. Studies have shown that HDACs are increased in synovial tissues of patients with RA and that inhibiting HDACs can reduce inflammation responses [113,114]. HDAC1 is a key player in arthritis mediated by the T cell [115]. Research carried out by our group has proven that HDAC6 increased in the synovium tissues of adjuvant-induced arthritic rats [116].

HDAC inhibitors have been shown to decrease the production of pro-inflammatory cytokines, such as TNF-α and IL-6, and to increase the activity of regulatory T cells, which can help to suppress the immune response in RA [117,118]. Several HDAC inhibitors, such as HDAC6 inhibitors CKD-506 [119] and CKD-L [120] and HDAC6 inhibitor NK-HDAC1 [121], are currently being investigated as potential treatments for RA in pre-clinical studies. Additionally, further research is needed to understand further the role of HDACs in RA pathogenesis and the potential benefits of HDAC inhibitors in the treatment of RA.

The hydrogen sulfide (H2S) donor S-propargyl cysteine (SPRC/ZYZ-802) developed by our team alleviates the inflammatory response and inhibits HDAC6 and JMJD3 in RA models [108,116]; it might be a potential agent for RA treatment in the future.

### 4.2. DNA Methylation

Compared with other epigenetic modifications, DNA methylation has the advantage of stabilizing inheritance [122]. DNA methylation is the binding of a methyl group at the cytosine 5 carbon site of CpG dinucleotide in the genome through the action of DNA methyltransferase without changing the DNA sequence. CpG islands are the most common regions where DNA methylation occurs. DNA methylation is induced by DNMTs DNA methyltransferases (DNMTs), the most important of which is DNMT1. There are differences in DNA methylation between Rheumatoid arthritis synovial fibroblasts (RASFs) and Osteoarthrosis synovial fibroblasts (OASFs). RASF is hypomethylated and DNMT1 expression is low [123]. As early as 1986, DNA methyltransferase inhibitor 5-azacytidine was found to induce autoimmune-related symptoms [124]. Studies have shown that in PBMCs of RA patients, CpG methylation in the promoter region of IL-6 can regulate the pathogenesis of RA [125]. CD4^+^ T cells of RA patients show hypomethylation, and CD4^+^CD8^+^ T cells have been found to have significantly hypomethylated IFNγ promoters, which can produce higher levels of IFNγ [126,127]. In a recent study, the DNA methylation of T cells in RA patients was significantly different between synovium and blood, which provided a strong support for the discovery of new therapeutic targets by targeting T cells [128]. In general, hypermethylation of gene promoter regions usually prevents transcription factor binding, leading to transcriptional repression. The hypomethylation of genes is associated with increased gene expression, and a study found that DNA methylation levels of the Peptidylarginine deiminase type4 (PADI4) were significantly lower in RA patients compared to healthy controls. PADI4 is involved in the citrullination of proteins, which is thought to be a key process and a non-MHC genetic risk factor in the development of RA. The hypo-methylation of the PADI4 gene promoter region may lead to the increased expression of the gene, which in turn could contribute to the development and progression of RA [129].

In the PBMC isolated from RA patients, and the DNMTs inhibitor 5′-AzaC increased the expression of anti-inflammatory cytokine IL-10 [130]. Therefore, DNMTs inhibitors have been proposed as potential RA drugs. However, the relationship between DNA methylation and the occurrence and development of RA is complex and needs further research and exploration.

### 4.3. Micro RNA

MicroRNAs occupy an important position in the modification of non-coding RNAs, and their special property is that they are genome encoded. There are differences in miRNA expression in RA patients and various cells associated with RA. For example, in peripheral blood monocytes from patients with RA, a variety of miRNAs were detected to be upregulated (miR-16, miR-103a, miR-132, miR-145, miR-146a, and miR-155), and also down-regulated (miR-21, miR-125b, and miR548a), these may be related to T cell homeostasis [131]. The abnormal expression of miR-17 and miR-146a were found to be related to the imbalance of Treg cells in peripheral blood T cells [132]. In RASFs, a large number of miRNAs are involved in the regulation of pathways (Wnt, NF-κB, JAK/STAT). There are few studies on miRNA regulation by oxidative stress in RA. Studies have shown that particulate-matter-induced RA can produce ROS and activate the MAPK signaling pathway, thus downregulating miR137 [133]. Therefore, it can be speculated whether targeting miR137 can mediate ROS, thus providing new ideas for the treatment of RA. Mechanisms of some miRNA regulating RA have been listed in Table 3.

Targeting miRNAs has also emerged as a potential therapeutic strategy for RA. For example, inhibition of miR-155 has been shown to upregulate FOXO3 and reduce inflammation and proliferation of FLS [134]. Similarly, targeting miR-146a has been shown to reduce the invasion and migration of FLSs via the miR-146a/GATA6 axis [135,136].

Changes in miRNA expression levels have been proposed as potential biomarkers for predicting the therapeutic response to RA treatments. Studies have shown that drug treatment can modulate the expression of miRNAs in clinical patients with rheumatoid arthritis (RA). For example, methotrexate (MTX), a commonly used disease-modifying anti-rheumatic drug (DMARD), miR-16, miR-132, miR-22, miR-155, and miR-146a described as MTX-treatment response biomarkers [137]. MTX has been shown to cause increased marrow adiposity and reduced trabecular bone volume; it has been reported that MTX upregulates miR-6315 after treatment, which might be a vital cause of bone and marrow fat formation [138]. Similarly, statistically significant differences between miR-26b, miR-29, miR-451, and miR-522 were observed after olokizumab therapy [139].

Further studies and patients are needed to fully elucidate the role of miRNAs in RA treatment and to develop miRNA-based therapies for this disease.

**Table 3 biomolecules-13-00766-t003:** Potential epigenetic targets in RA.

Targets	Effects	References
JMJD3	Regulating the proliferation and migration ability of FLS through the AKT signaling pathway Regulating symptoms of CIA mice	[107,108]
GATA4	Mediating the proliferation and migration ability of FLS through MAPK signaling pathway Promoting angiogenesis	[109]
SIRT3	Regulating of ROS levels in miceInducing osteoarthritis in mice Inducing osteoarthritis in mice	[110,111]
SIRT4	Regulating the expression of SOD1, SOD2, and catalase Targeting osteoarthritis with inflammatory factors	[112]
HDAC1	Knockdown relieves joint swelling and synovial inflammation in arthritic mice	[115]
HDAC6	Increase in the synovium tissues of adjuvant-induced arthritic rats	[116]
miR-137	Decreasing expression is due to ROS activation of MAPK in AIA rats	[133]
miR-155	Upregulating FOXO3 and reducing inflammation and proliferation of FLS	[134]
miR-146a	Targeting miR-146a reduces the invasion and migration of FLSs via the miR-146a/GATA6 axis	[135]
NAV2	Mediating the proliferation and migration of MH7A via Wnt/β-catenin signaling pathway	[140,141]
miR-203	Promoting the production of MMP1 and IL-6 induces RA	[142]
miR-19	Inducing inflammation through regulation of TLR2, IL-6, and MMP3 in FLS stimulated by TNFα Having an anti-inflammatory effect in FLS stimulated by LPS by regulating IL-6 and IL-1β	[143,144,145]
miR-10a	Mediating the production of inflammatory factors through NF-κB in RA-FLS	[146]
miR-124a	Reducing the proliferation of synovial cells Relieving cartilage or bone destruction and the symptoms in AIA rats	[147]
miR-152	Upregulating the expression of SFRP4 (negative regulator of Wnt signaling pathway) by targeting DNMT1, and reducing the proliferation of FLSs by inhibiting the activation of Wnt pathway	[148]
miR-375	Decreasing the expression of FZD8 (a member of Wnt pathway), inhibiting the expression of IL-6 and IL-8, and relieving the symptoms of AIA rats	[149]

## 5. Conclusions and Perspective

RA, as a systemic inflammatory disease, is not fatal, but it harms patients’ physical and mental health. At present, there are a variety of therapeutic drugs and methods in clinical practice, but some of them have side effects, and about 50% of patients are not sensitive to the existing therapeutic targets, so it is difficult to achieve the ideal therapeutic effect. There are many factors in the development of RA, involving a variety of cells, cytokines, and signaling pathways. So, there is a great possibility of finding a better target.

In recent years, epigenetics has developed rapidly, providing a lot of ideas for finding new targets for RA treatment. With the rapid development of bioinformatics, the application of multiomics analysis has become more extensive and we can find potential new targets more efficiently. In recent decades, the rise in biological targeted drugs has provided multiple options for the treatment of RA. Although there are still many targets that have not been applied clinically, we believe that in the future, targeted drugs will provide better treatment, less cost, faster therapeutic effects, and fewer side effects for a large number of RA patients.

Our research group proved for the first time that Neuron Navigator 2 (NAV2) promotes the inflammatory response in RA, and we noticed that the transcription factor E2F1 can bind to the NAV2 promoter region to activate the transcription and expression of NAV2 [54,140,141]. Previous studies have demonstrated that NAV2 plays a key role in the development of the mammalian nervous system [150], and we thus propose that NAV2 may affect inflammatory pain during RA disease progression through neuronal navigation [151], which may provide a basis for alleviating the pain of RA patients. At the same time, we also found that the level of GATA4 increased in the synovium of RA patients. The transcription factor GATA4 is an important regulator of cardiac-differentiation-specific gene expression. We demonstrated for the first time that GATA4 plays a role in regulating VEGF in RA FLS, inducing cell migration/proliferation, and plays an important role in angiogenesis [109]. Our study also showed that the protein level of histone deacetylase HDAC6 was increased in the synovial tissue of arthritic rats [116]. In an animal model of RA, HDAC inhibitors could improve synovial inflammation and joint swelling and relieve RA symptoms [117,118]. At the same time, we also found that the expression of histone demethylase JMJD3 was increased in PDGF-induced FLS, after inhibition or silence of JMJD3, the migration, and proliferation of FLS receded [107].

The hydrogen sulfide (H2S) donor S-propargyl cysteine (SPRC/ZYZ-802) independently developed by our team alleviates the inflammatory response through the Nrf2-ARE signaling pathway [152], or the HDAC6/MyD88/NF-κB signaling pathway inhibited the expression of HDAC6 [116]. Meanwhile, we also found that CSE/H2S can reduce the expression of JMJD3 and reduce arthritis. These findings proposed that SPRC may be used as a potential drug for the treatment of rheumatoid arthritis. We have developed two types of H2S sustained-release donors [153,154] that solve the problem of the too-fast release of H2S in traditional formulations. ZYZ-802 is currently being submitted to the CFDA and FDA for clinical trial applications [15].

In conclusion, the study of epigenetics has opened up new avenues for the development of therapeutics for rheumatoid arthritis (RA). Epigenetic modifications have been shown to play a crucial role in the development and progression of RA, and drugs that target these modifications have shown promising results in animal models of the disease.

Furthermore, epigenetic changes are reversible, which means that epigenetic therapies have the potential to not only treat the symptoms of RA but also modify the disease course and prevent its progression. This contrasts to conventional RA therapies, which are often palliative and do not modify the underlying disease process.

However, there are also challenges that need to be addressed before epigenetic therapies can be widely used in the clinic. One of the major challenges is the need to identify specific epigenetic modifications that are involved in the pathogenesis of RA and to develop drugs that target these modifications with high specificity and efficacy.

HDAC inhibitors and DNA methyltransferase inhibitors are two types of epigenetic drugs that have been studied extensively for the treatment of RA. While these drugs have shown some efficacy, there is still a need for further research to determine their safety and effectiveness in humans.

MicroRNAs have also emerged as potential targets for RA therapy. These small non-coding RNAs can regulate gene expression and have been shown to be involved in the pathogenesis of RA. MicroRNAs have gained significant attention as potential therapeutic targets and diagnostic biomarkers for RA.

In addition to the potential use of epigenetic drugs, there is also growing interest in the role of the microbiome in the development of RA. Studies have suggested that changes in the gut microbiome could play a role in the development of RA, and targeting the microbiome with probiotics or fecal microbiota transplantation could be a potential therapeutic strategy for the disease [155,156].

Another area of research is the use of gene editing technologies, such as CRISPR-Cas9 [157,158,159], to modify the expression of specific genes involved in the pathogenesis of RA. While this approach is still in its early stages, it holds promise for the development of highly targeted and personalized therapies for RA.

Overall, although there is currently no cure for RA, effective treatment options are available that can help manage the symptoms and improve the quality of life for people with this condition. Drug development of new targets is expected to provide more options for patients who do not respond to current treatments. Ongoing research into the underlying causes of RA may lead to new and more effective treatments in the future. The study of epigenetics has provided new insights into the complex pathogenesis of RA and has opened up new possibilities for the development of targeted therapies. While more research is needed, the use of potential therapeutics for the treatment of RA holds great promise for improving the lives of millions of people affected by this chronic autoimmune disorder.

## Figures and Tables

**Figure 1 biomolecules-13-00766-f001:**
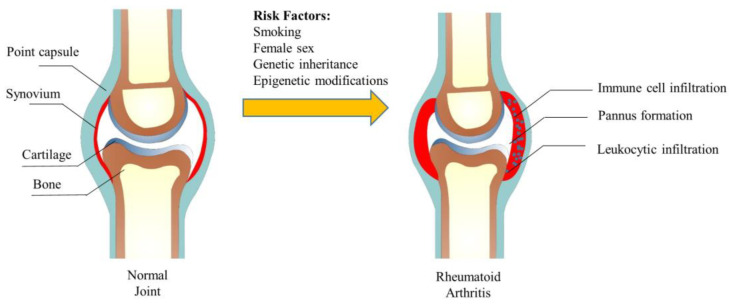
The structure of the joint and its change in rheumatoid arthritis.

**Figure 2 biomolecules-13-00766-f002:**
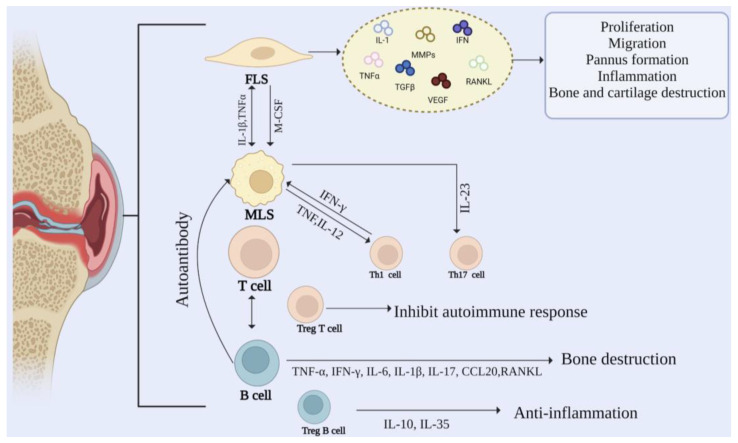
Cellular mechanisms in rheumatoid arthritis. The pathogenesis of RA is regulated by the interaction between FLS, MLS, and immune cells.

**Figure 3 biomolecules-13-00766-f003:**
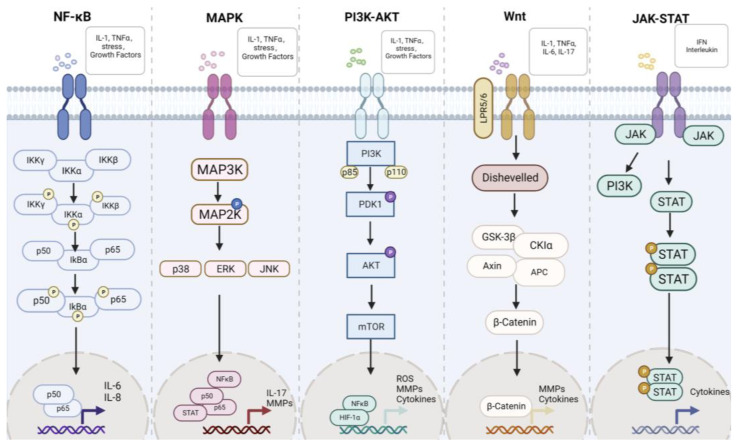
The main signaling pathway in rheumatoid arthritis. NF-κB, MAPK, PI3K-AKT, Wnt, and JAK-STAT are the classic pathways in RA.

**Figure 4 biomolecules-13-00766-f004:**
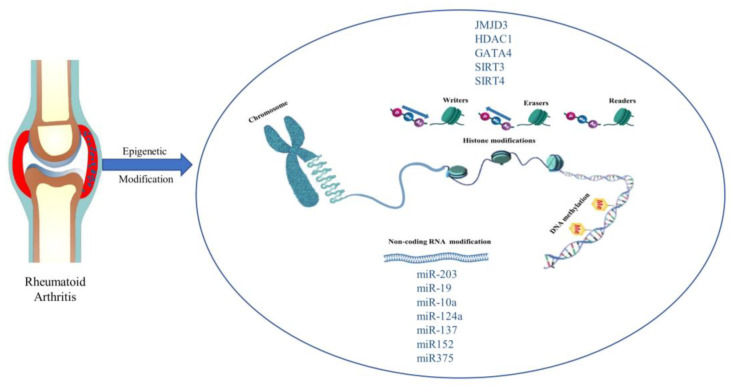
Epigenetic modification and promising new targets in rheumatoid arthritis. Histone modifications, DNA methylation, and Non-coding RNA modifications are common in epigenetic studies of RA.

**Table 1 biomolecules-13-00766-t001:** Biological agents for the treatment of RA.

Biological Agents	Target Cell	Target Factor	Mechanisms	Status	References
Etanercept (ENT)		TNF-α	In binding to TNFα and TNFβ, it has a more remarkable ability than endogenous soluble TNFRs to prevent the proinflammatory reaction	Approved	[66]
Infliximab (IFX)		TNF-α	It binds to the target receptor of TNFα and inhibits the production of inflammatory factors	Approved	[67,68]
Adalimumab (ADA)		TNF-α	It specifically binds to TNFα and blocks its interaction with TNF receptors on p55 and p75 cells	Approved	[69,70]
Golimumab		TNF-α	Like Infliximab, it has a strong binding force on soluble and transmembrane TNFα, resulting in TNFα neutralization	Approved	[71]
Certolizumab pegol		TNF-α	It is a human antibody consisting of an antigen-binding fragment (Fab ‘) of an anti-TNF-α monoclonal antibody	Approved	[72]
Rituximab	B cells		It removes all B cells except the pre-B cells and plasma cells	Approved	[73]
Abatacept (CTLA4-Ig)	T cells		By binding to CD80 and CD86 on the surface of B cells, it inhibits co-stimulation and activation of T cells, leading to the downregulation of inflammatory mediators	Approved	[74]
Tocilizumab		IL-6	It inhibits IL-6-mediated inflammatory responses, induces/expanses Breg cells, and down-regulates the expression of pro-inflammatory factors.	Approved	[75]
Sirukumab		IL-6	It binds IL-6R and blocks its binding with IL-6 molecules to inhibit the IL-6 signaling pathway to slow the disease	Approved	[76]
Olokizumab		IL-6	It recognizes the “site 3” site of IL-6, thus preventing the binding of IL-6 to gp130 and achieving the purpose of inhibiting the IL-6 pathway	Approved	[77]
Clazakizumab		IL-6	It targets soluble and membrane-bound IL-6 receptors to prevent IL-6 receptor complex formation and inhibit IL-6 signaling	Approved	[78]
Brodalumab		IL-17	It leads to the disruption of the IL-17 pathway by blocking the activities of 17A, 17E, and 17F	Approved	[79]
Tofacitinib		JAK	It inhibits STAT phosphorylation and activation through JAK1/JAK3 dimer-mediated signaling and inhibits B lymphocyte activation and differentiation	Approved	[80]
Baricitinib		JAK	It selectively inhibits JAK1 and JAK2	Approved	[81]
Upadacitinib		JAK	It selectively inhibits JAK1 and effectively inhibits STAT3 phosphorylation induced by IL-6 and IFNγ	Approved	[82]
Filgotinib		JAK	It causes the selective inhibition of JAK1 and reduces the pro-inflammatory response	Approved	[83]
Peficitinib		JAK	It is a JAK3 inhibitor, restraining the activation of pro-inflammatory cytokine signaling	Approved	[84]
SHR0302		JAK	Its highly selective inhibition of JAK1 suppresses the inflammatory response	Clinical trial phase 3 (NCT04333771)	[85]
VX-509		JAK	It causes selective JAK3 inhibition and mitigates the inflammatory response	Clinical trial phase 2/3 (NCT01830985)	[86]

**Table 2 biomolecules-13-00766-t002:** Natural compounds for potential treatment of RA.

Extract	Source	In Vivo	In Vitro	Potential Pathway	References
Rhoifolin	A flavanone extracted from Rhus succedanea	Alleviating the symptoms of CIA mice, and improving cytokines and oxidation indexes		NF-κB	[87]
Perillyl alcohol	A monoterpene separated and collected from the essential oils of citronella, lavandin, peppermint, spearmint, celery seeds, and a few different plants	Improving the levels of MDA, GSH, NO, IL-1β, IL-10, IL-6, TNFα, and PGE2 in CIA mice	Protecting LPS-induced RAW 264.7 activation and improving inflammatory and oxidative markers	TLR4/NF-κB and Keap1/Nrf2	[88]
Resveratrol	A natural polyphenol found in grapes and Polygonum cuspidatum	Attenuating paw swelling and decreasing serum antioxidant enzyme levels in AA rats	Activating antioxidant pathways and inhibiting the proliferation of FLS	Nrf2-ARE	[89]
			Reducing H_2_O_2_-induced mitochondrial ROS production	Nrf2–Keap1	[90]
Salvianolic acid	A major phytoconstituent of the plant Radix Salvia miltiorrhiza.	Alleviating oxidative stress state in mice and antagonizes free radicals			[91]
wogonin	A flavonoid contained in the Scutellaria Baicalensis Georgi root	Decreasing the content of lipid peroxides in CFA rats			[92]
Magnoflorine	A main component purified from Clematis manshurica Rupr.	Decreasing inflammatory responses in AIA rats	Decreasing the proliferation, migration, invasion, and reactive oxygen species levels in IL-1β-treated MH7A	PI3K/Akt/NF-κBKeap1-Nrf2/HO-1	[93]
Chelerythrine	A benzo phenanthridine alkaloid isolated from the root of *Papaveraceae*.	Protecting AIA rats from inflammation and bone damage.	Inhibiting the migration and colony-formation of the HFLS-RA, increasing the intracellular level of ROS, promoting apoptosis	AMPK/mTOR/ULK-1	[94]
Cantleyoside	A iridoid glycosides in *P. hookeri*		Leading to mitochondrial damage by Ca^2+^ overload and ROS release, promotes apoptosis and inhibits proliferation of HFLS-RA	AMPK/Sirt 1/NF-κB	[95]
Icariin	A prenylated flavonol glycoside isolated from *Epimedium*		Interfering with cell cycle progression, increasing intracellular ROS level, inducing cell apoptosis, and inhibiting the proliferation of FLS		[96]
Neohesperidin	A flavone compound isolated from various dietary sources		Anti-migration, anti-invasion, anti-oxidative, and apoptosis-induced in RA-FLS	MAPK	[97]
Hesperidin	A flavanone present in large quantities in citrus fruits	Reducing inflammation in CIA rats, regulating adenosine nucleotide and nucleoside hydrolases, reducing intracellular ROS, attenuating the apoptotic process of bone marrow cells			[98]
Celastrol	A quinone-methylated triterpenoid extracted from *Tripterygium wilfordii*	Reducing the degree of joint swelling, arthritis index score, inflammatory cell infiltration and synovial hyperplasia	Inhibiting the activation of NLRP3 inflammasome and the production of ROS induced by LPS and ATP in THP-1 cells.		[99]
		Reducing the expression of inflammatory factors in the serum of CIA-induced RA mice, alleviate cartilage tissue damage and inflammatory invasion in mice	Inhibiting FLSs proliferation and enhancing autophagy	PI3K/AKT/mTOR	[100]
Fangchinoline	An alkaloid can be extracted from the roots of *Stephaniae tetrandra*	Improving behavioral parameters and inflammatory signs in C/K rats and CIA mouse	Decreasing the production of inflammatory cytokines and ROS in HFLS	MAPK NF-κB	[101]
Sinomenine		Alleviating symptoms in CAIA mice by protecting their joints from destruction	Inhibiting the secretion of IL-6 and IL-33 and ROS production in RASFs, thereby mediating the protective effect against bone destruction	Keap1-Nrf2	[102]
Gentiolic acid	An active component in the whole plant or root of *Gaultheria leucocarpa Bl.va. crenulata (Kurz) t.z.Hsu*	Reducing the expression of inflammatory factors in serum of CIA rats and relieving the symptoms of RA	Inhibiting proliferation and invasion of MH7A, and the apoptosis was promoted	RAF/ERK	[103]
Curcumin	A tetraterpenoid obtained from *Curcuma longa*	Inhibiting the expression of pro-inflammatory cytokines TNF-α, IL-6, and IL-17 in CIA-stimulated mice, relieving hind paw edema in RA mice	Reducing TNFα-induced MH7A proliferation and migration ability and promoting cell apoptosis.	PI3K/AKT	[104]

## Data Availability

Not applicable.

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
