# Peer review of "New Targets and Strategies for Rheumatoid Arthritis: From Signal Transduction to Epigenetic Aspect"

_biomolecules, 2023, doi:10.3390/biom13050766_

Round 1

Reviewer 1 Report

The manuscript by Zhu et al. represents a review article, which summarizes current knowledge on some potential RA targets discovered from the perspective of epigenetics and RA mechanisms. While the topic of this review is timely and interesting, in my opinion, some improvements should be performed prior the possible acceptance of this manuscript for the publication. 

Major

-  a big set of the information on the ongoing and recently finished clinical trials of novel compounds for RA treatment are missed, while this information shows current tendencies in the field; 

- the literature contains some information linking epigenetic factors with therapeutics available (for example see PMID: 33142700; PMID: 36361779). In my opinion, additional chapter discussing extensively these works could be a good link connecting ‘therapeutics’ and ‘epigenetics’ parts of the review;   

- Conclusions are very broad and unspecific. It is very interesting to know authors’ opinion on the most perspective area development directions.   

Minor 

- title to the table 2: natural compounds instead of extracts should be used;

- due to the plenty of free space within the figures, the fonts could be enlarged.   

Author Response

Response to Reviewer 1 Comments

The manuscript by Zhu et al. represents a review article, which summarizes current knowledge on some potential RA targets discovered from the perspective of epigenetics and RA mechanisms. While the topic of this review is timely and interesting, in my opinion, some improvements should be performed prior the possible acceptance of this manuscript for the publication. 

Major

Point 1a big set of the information on the ongoing and recently finished clinical trials of novel compounds for RA treatment are missed, while this information shows current tendencies in the field; 

Response 1: That is true, thanks for your advice, we've replenished the ongoing and recently finished clinical trials of some promising novel compounds for RA treatment. JAK inhibitors are still the current tendencies in the field.

Point 2the literature contains some information linking epigenetic factors with therapeutics available (for example see PMID: 33142700; PMID: 36361779). In my opinion, additional chapter discussing extensively these works could be a good link connecting ‘therapeutics’ and ‘epigenetics’ parts of the review;   

Response 2: Thanks for your very good suggestion, we have added the corresponding content of microRNA as the therapeutic response for RA treatment.

.

Point 3Conclusions are very broad and unspecific. It is very interesting to know authors’ opinion on the most perspective area development directions.   

Response 3: Yes, the conclusion has been expanded and highlighted our research and opinion.

Minor 

Point 4title to the table 2: natural compounds instead of extracts should be used;

Response 4: Replaced, thanks

Point 5 due to the plenty of free space within the figures, the fonts could be enlarged

Response 5: We've made adjustments,thanks

Reviewer 2 Report

The authors have summarized the pathogenesis of rheumatoid arthritis and current treatment approaches. However, the sections covering epigenetic events is rather short. Often, the sentences are not structured and misleading. Further, the English of the manuscript needs to be improved. I can only name a few examples in this context:

-          25: “There is only a small amount of….” This sentence is not introduced appropriately and confuses the reader.

-          31: “Since diet plays…” This sentence is rather long and difficult to understand.

-          35: “The healthy synovial…” Then authors switch again to synovial tissue, which makes the entire introduction unstructured.

-          41: “The neutrophils…” Again this sentence is not related to the sentence before or after, this is confusing for the reader.

-          59: “…leading to dysfunction of synovial tissue” This wording sounds strange, please rephrase

-          60 + 63 “.. the treatment of RA…” this sentence is written twice

-          67: “excessive” might be the wrong word in this context

-          70: “proliferation and differentiation of these cells” What cells do the authors mean, please clarify

-          73: “DRAMDs” is a typo

-          99: The abbreviation “ECM” is not mentioned

-          127: “the important role of cytokines…” Please mention references here.

-          142: “is highly expressive of miR-155” This wording is not scientific.

-          156: “In addition, B cells are related to the production” B cells are not only related, they produce the antibodies. Please rephrase.

-          171: “inflammatory factors” Which inflammatory factors do the authors mean?

-          178: Where is the connection to Hydrogen in this paragraph? Please specify.

-          184-185: If the Wnt pathway is increased in RA patients, how can an inhibitor, decreasing Wnt signaling, cause pannus formation? Please clarify this.

-          198: “pathway is about to be turned on” Please use scientific writing.

-          216-234: This whole paragraph is complicated and unstructured. It is very difficult for the reader to understand.

-          250: “An increasing amount…” How can more and more enzymes act as writers, do they decide it for themselves? Do the authors mean that more and more are discovered?

-          269: “mainly in proteins..” Did the authors mean that histone methylases contain SET domains? This is not clear from the sentence…

-          279: “This conclusion may provide ideas…” This conclusion is not clear to me, the authors need to explain this more explicitly.

-          283: Can SOD decrease alone cause osteoarthritis in mice – please rephrase or state correctly

-          296: RASF and OASF: Please explain abbreviation and function

-          312: “This may be related to T cell…” This sentence is without any context, and no reference is mentioned.

Author Response

Response to Reviewer 2 Comments

Point: The authors have summarized the pathogenesis of rheumatoid arthritis and current treatment approaches. However, the sections covering epigenetic events is rather short. Often, the sentences are not structured and misleading. Further, the English of the manuscript needs to be improved. I can only name a few examples in this context:

-          25: “There is only a small amount of….” This sentence is not introduced appropriately and confuses the reader.

-          31: “Since diet plays…” This sentence is rather long and difficult to understand.

-          35: “The healthy synovial…” Then authors switch again to synovial tissue, which makes the entire introduction unstructured.

-          41: “The neutrophils…” Again this sentence is not related to the sentence before or after, this is confusing for the reader.

-          59: “…leading to dysfunction of synovial tissue” This wording sounds strange, please rephrase

-          60 + 63 “.. the treatment of RA…” this sentence is written twice

-          67: “excessive” might be the wrong word in this context

-          70: “proliferation and differentiation of these cells” What cells do the authors mean, please clarify

-          73: “DRAMDs” is a typo

-          99: The abbreviation “ECM” is not mentioned

-          127: “the important role of cytokines…” Please mention references here.

-          142: “is highly expressive of miR-155” This wording is not scientific.

-          156: “In addition, B cells are related to the production” B cells are not only related, they produce the antibodies. Please rephrase.

-          171: “inflammatory factors” Which inflammatory factors do the authors mean?

-          178: Where is the connection to Hydrogen in this paragraph? Please specify.

-          184-185: If the Wnt pathway is increased in RA patients, how can an inhibitor, decreasing Wnt signaling, cause pannus formation? Please clarify this.

-          198: “pathway is about to be turned on” Please use scientific writing.

-          216-234: This whole paragraph is complicated and unstructured. It is very difficult for the reader to understand.

-          250: “An increasing amount…” How can more and more enzymes act as writers, do they decide it for themselves? Do the authors mean that more and more are discovered?

-          269: “mainly in proteins..” Did the authors mean that histone methylases contain SET domains? This is not clear from the sentence…

-          279: “This conclusion may provide ideas…” This conclusion is not clear to me, the authors need to explain this more explicitly.

-          283: Can SOD decrease alone cause osteoarthritis in mice – please rephrase or state correctly

-          296: RASF and OASF: Please explain abbreviation and function

-          312: “This may be related to T cell…” This sentence is without any context, and no reference is mentioned.

Response: Thank you for your detailed revision suggestion. We have revised the title and original text in corresponding places to make the expression more clear and make the logic more coherent.

Reviewer 3 Report

This review by Zhu et al. aims to describe new molecules that could be targeted for treatment of Rheumatoid Arthritis (RA), focusing on epigenetic factors.

As most of the current treatments for RA are meant to relieve pain, a list of new targets that could potentially be used as a cure is very interesting.  The review provides some ideas for new potential targets but they are often hidden in a lot of unnecessary detail and it is often hard to understand what targets they are predicting could be therapeutic and why.  It is also hard to tell if they predict each target could be used as a better pain reliever or if they think they could be potential cures. Finally, while the title suggests a focus on epigenetic molecules, this is only a small part of the review.

I suggest acceptance of this paper with the following minor revisions.

1.     Reorganize by potential targets. Each target should be its own heading and they could further be divided in sections for targets predicted to relieve pain vs targets that could be curative.

2.     Either change the title or make the epigenetics section more central.  Currently, it is at the end and fairly short, so it is unclear why it is the focus of the title.

Author Response

Response to Reviewer 3 Comments

This review by Zhu et al. aims to describe new molecules that could be targeted for treatment of Rheumatoid Arthritis (RA), focusing on epigenetic factors.

As most of the current treatments for RA are meant to relieve pain, a list of new targets that could potentially be used as a cure is very interesting.  The review provides some ideas for new potential targets but they are often hidden in a lot of unnecessary detail and it is often hard to understand what targets they are predicting could be therapeutic and why.  It is also hard to tell if they predict each target could be used as a better pain reliever or if they think they could be potential cures. Finally, while the title suggests a focus on epigenetic molecules, this is only a small part of the review.

I suggest acceptance of this paper with the following minor revisions.

Point 1: Reorganize by potential targets. Each target should be its own heading and they could further be divided in sections for targets predicted to relieve pain vs targets that could be curative.

Response 1: Thanks for your suggestion, Currently, there is no known cure for rheumatoid arthritis. However, with appropriate treatment, many people with RA can achieve remission or a significant reduction in symptoms. Drug development of new targets is expected to provide more options for patients who do not respond to current treatments. We have revised the text and discussed potential new targets for therapy based on our research in the conclusion.

Point 2: Either change the title or make the epigenetics section more central.  Currently, it is at the end and fairly short, so it is unclear why it is the focus of the title.

Response 2: Thanks for your suggestion, we have revised the title.

Round 2

Reviewer 1 Report

I appreciate the authors ‘work making improvements.

Author Response

Response to Reviewer 1 Comments

Point 1I appreciate the authors ‘work making improvements.

Response 1: Thanks for your comment.

Reviewer 2 Report

The authors managed to improve the manuscript significantly. However, still, some minor grammar issues and typos can be found in the revised version. Please correct it, then the manuscript is ready for publication.

Author Response

Response to Reviewer 2 Comments

Point: The authors managed to improve the manuscript significantly. However, still, some minor grammar issues and typos can be found in the revised version. Please correct it, then the manuscript is ready for publication.

Response: Thanks for your suggestion. We have revised the grammar issues and typos in the corresponding places in original text.
